# Phytochemical Combination (*p*-Synephrine, *p*-Octopamine Hydrochloride, and Hispidulin) for Improving Obesity in Obese Mice Induced by High-Fat Diet

**DOI:** 10.3390/nu14102164

**Published:** 2022-05-23

**Authors:** Dahae Lee, Ji Hwan Lee, Byoung Ha Kim, Sanghyun Lee, Dong-Wook Kim, Ki Sung Kang

**Affiliations:** 1College of Korean Medicine, Gachon University, Seongnam 13120, Korea; pjsldh@gachon.ac.kr (D.L.); kleert26@naver.com (J.H.L.); 2D. Nature Co., Ltd., Seongnam 13174, Korea; mot37@d-nature.co.kr; 3Department of Plant Science and Technology, Chung-Ang University, Anseong 17546, Korea; slee@cau.ac.kr; 4College of Pharmacy, Wonkwang University, Iksan 54538, Korea

**Keywords:** hispidulin, *p*-synephrine, *p*-octopamine hydrochloride, adipocytes, adipogenesis

## Abstract

Obesity treatment efficiency can be increased by targeting both central and peripheral pathways. In a previous study, we identified two natural compounds (hispidulin and *p*-synephrine) that affect adipocyte differentiation. We tested whether obesity treatment efficiency may be improved by adding an appetite-controlling agent to the treatment in the present study. Alkaloids, such as *p*-octopamine, are adrenergic agonists and are thus used as dietary supplements to achieve weight loss. Here, we assessed anti-obesity effects of a mixture of *p*-synephrine, *p*-octopamine HCl, and hispidulin (SOH) on murine preadipocyte cells and on mice receiving a high-fat diet (HFD)**.** SOH showed stronger inhibition of the formation of red-stained lipid droplets than co-treatment with hispidulin and *p*-synephrine. Moreover, SOH reduced the expression of adipogenic marker proteins, including CCAAT/enhancer-binding protein alpha, CCAAT/enhancer-binding protein beta, and peroxisome proliferator-activated receptor gamma. In the HFD-induced obesity model, body weight and dietary intake were lower in mice treated with SOH than in the controls. Additionally, liver weight and the levels of alanine aminotransferase and total cholesterol were lower in SOH-treated mice than in the controls. In conclusion, our results suggest that consumption of SOH may be a potential alternative strategy to counteract obesity.

## 1. Introduction

Obesity is the underlying cause of various metabolic disorders such as hypertension, hyperglycemia, and diabetes. Therefore, it has become a global medical issue [1]. Imbalances between energy intake and consumption can lead to excessive fat accumulation and functional and morphological changes in adipocytes, which is characteristic of obesity [2,3]. Typically, obesity develops due to an increased number and size of adipocytes. Therefore, controlling the differentiation of preadipocytes into mature adipocytes (termed adipogenesis) is important in the treatment of obesity [4]. Although the precise molecular mechanisms underlying obesity remain unclear, recent studies indicated that CCAAT/enhancer-binding protein alpha (C/EBPα), C/EBPβ, and peroxisome proliferator-activated receptor gamma (PPARγ) are important regulators released during adipogenesis [5,6].

Several approaches such as inhibition of dietary fat absorption, appetite control, insulin and leptin revival, inhibition of fat synthesis, and increased fat mobilization and burning have been used to develop obesity treatments. The FDA-approved combination drug Qsymia^®^, which is a combination of the stimulant phentermine and the anti-seizure drug topiramate, highlights the synergistic effect of the individual compounds. As this drug is associated with undesirable side effects, such as increased blood pressure, anxiety, and dizziness, recent studies attempted to develop safe anti-obesity drugs derived from natural products [7,8]. Of note, compounds isolated from natural products for the prevention and treatment of obesity are frequently used as mixtures because such mixtures may exert synergistic effects as a result of different mechanisms of action [9]. For instance, combined treatment with naringin, hesperidin, and *p*-synephrine results in weight loss in human patients [10]. In our previous study, combinations of phytochemicals (*p*-synephrine and hispidulin) produced anti-adipogenic effects in a murine preadipocyte cell line (3T3-L1), and *p*-synephrine and hispidulin exhibited different mechanisms of action [11]. Several studies confirmed low or negligible toxicity of these compounds in animal models [12,13]; however, the anti-obesity effects of co-treatment with *p*-synephrine and hispidulin in animal models are still unclear. Hispidulin has been isolated from *Grindelia argentina* and *Arrabidaea chica* [14]. *p*-Synephrine is abundant in *Citrus aurantium*, and it is biosynthesized through N-methylation of *p*-octopamine [15,16].

We reviewed the literature and found that obesity treatment efficiency may be improved by synchronously targeting central and peripheral pathways using natural products. We hypothesized that *p*-octopamine was a suitable natural ingredient that may exert effects similar to those of Qsymia phentermine. Alkaloids, such as *p*-octopamine, exhibit adrenergic agonist activity and are used as dietary supplements to achieve weight loss. *p*-Octopamine is a hydroxylated phenylethylamine that occurs naturally in animals, including insects. In particular, *p*-octopamine is abundant in *Citrus limon* and *Citrus aurantium*. In obese Zucker rats, intraperitoneal administration of *p*-octopamine results in weight loss without adverse effects [17]. *p*-Octopamine hydrochloride (*p*-octopamine HCl) is used for the treatment of circulatory disorders and hypotensive regulation [15,18,19]; however, its potential anti-obesity effects remain unclear. Here, we assessed antidiabetic effects of a mixture of *p*-synephrine, *p*-octopamine HCl, and hispidulin (SOH) in 3T3-L1 adipocytes and high-fat diet (HFD)-fed mice. We hypothesized that SOH treatment would significantly reduce adipogenesis in 3T3-L1 adipocytes and reduce the body weight of HFD-fed mice.

## 2. Materials and Methods

### 2.1. Cell Culture and Adipogenic Differentiation

3T3-L1 mouse preadipocyte cells were obtained from the American Type Culture Collection (Manassas, VA, USA). The 3T3-L1 cells were cultured in Dulbecco’s modified Eagle’s medium (Cellgro, Manassas, VA, USA) containing antibiotics, i.e., 1% penicillin/streptomycin (P/S; Gaithersburg, MD, USA), and 10% bovine calf serum (Gaithersburg, MD, USA). The 3T3-L1 preadipocytes were seeded in a 24-well plate at 4 × 10^4^ cells/well, and after two days, the medium was replaced by adipogenic differentiation medium containing 1-methyl-3-isobutylxanthine (Sigma-Aldrich, St. Louis, MO, USA), 1% P/S, 0.4 μg/mL dexamethasone (Sigma-Aldrich), 10% fetal bovine serum (FBS; Gaithersburg), and 5 µg/mL insulin (Sigma-Aldrich). After incubation for two days, the cultivation medium was replaced by adipogenic differentiation medium containing 1% P/S, 10% FBS, and 5 µg/mL insulin, which was exchanged every two days. On differentiation day 6, the medium was replaced with adipogenic differentiation medium containing 10% FBS and 1% P/S, followed by cultivation for two days. Hispidulin, *p*-synephrine, and *p*-octopamine HCl were added individually or in combination to the culture medium during adipogenic differentiation. *p*-Synephrine (≥98%), and *p*-octopamine HCl (≥95%) were purchased from Sigma-Aldrich. Hispidulin (≥98%) was obtained from Natural Product Institute of Science and Technology (www.nist.re.kr), Anseong, Korea.

### 2.2. Cell Viability

Viability of 3T3-L1 cells was measured using a water-soluble tetrazolium salt-1-based colorimetric assay kit (Ez-Cytox Cell Viability Assay Kit; Daeil Lab Service, Seoul, South Korea). 3T3-L1 cells were seeded in a 96-well plate at 1 × 10^4^ cells/well. Hispidulin, *p*-synephrine, and/or *p*-octopamine HCl were then individually added at different concentrations (5, 10, 20, or 40 μM, each) for 8 days, 10 μL EZ-Cytox reagent was added, and absorbance was measured after 40 min using a PowerWave XS microplate reader (Bio-Tek Instruments, Winooski, VT, USA) at 490 nm.

### 2.3. Oil Red O Staining

After cell differentiation, differentiated cells were fixed using 4% paraformaldehyde solution (Sigma-Aldrich) for 1 h. Fixed cells were stained using Oil Red O solution (ORO; Sigma-Aldrich) for 1 h. After removal of the ORO solution, the cells were washed using distilled water, and images were recorded. Accumulated intracellular ORO was completely eluted using 100% isopropanol and was quantified through spectrophotometric absorbance using a PowerWave XS microplate reader (Bio-Tek Instruments) at 540 nm.

### 2.4. Western Blotting Analysis

Equal amounts of cellular lysates were evaluated by 10% sodium dodecyl sulfate–polyacrylamide gel and transferred by electroblotting to polyvinylidene difluoride membranes (Pall Corporation, Washington, DC, USA). The proteins were incubated with the primary antibodies, including PPARγ (53 kDa), C/EBPα (42 kDa), C/EBPβ (46 kDa), and GAPDH (37 kDa), overnight at 4 °C, and then incubated with an appropriate secondary antibody for 1 h at room temperature. All antibodies were purchased from Cell Signaling (Boston, MA, USA). Immune complexes were developed with enhanced chemiluminescence reagent (GE Healthcare UK Limited, Buckinghamshire, UK). Reactions were visualized using a chemiluminescence system (FUSION Solo; PEQLAB Biotechnologie GmbH, Erlangen, Germany).

### 2.5. Study Animals and Experimental Design

Male C57/BL6 mice (five weeks old upon receipt, Daehan Biolink, Chungbuk, Korea) were used for the experiments, after one week of acclimation. During acclimation and experimental periods, four mice were kept per cage at constant temperature (20–25 °C) and humidity (30–35%) and under a 12/12-h light/dark cycle. The mice had ad libitum access to feed and water. Four treatment groups (comprising eight individuals each) were used: (1) normal diet control group (ND); ND and administration of sodium carboxyl methyl cellulose (CMC) solution. (2) HFD control group: HFD after administration of the CMC solution; (3) positive group: HFD and administration of atorvastatin (ATS; 30 mg/kg); and (4) experimental group: HFD and administration of SOH (i.e., a 1:1:1 mixture of hispidulin, synephrine, and *p*-octopamine HCl at 20 mg/kg). The high fat diet (HFD) used food with 60% of energy from lipids, 18% of energy from protein and 21% of energy from carbohydrate (D12492 Research Diets, New Brunswick, NJ, USA). The food intake and body weight during the experimental period were measured weekly. All treatment compounds were dissolved in CMC solution and were orally administered at 5 mL/kg body weight. The blood was collected after the animals were sacrificed at the end of the experiment. The plasma was separated from collected blood and stored at −20 °C until analysis. The study was approved by institutional animal care and use committees of Gachon University (GU1-2022-IA0027-00). The experiment procedures were conducted based on the guidelines of the Declaration of Helsinki.

### 2.6. Blood Biochemistry

At the end of the experimental period, blood samples were collected from the inferior vena cava after fasting for 24 h and were placed in test tubes containing 0.18 M ethylenediaminetetraacetic acid (EDTA). The plasma was centrifuged in collected blood for 15 min at 1200 RCF and then stored at −20 °C until analysis. The levels of creatinine, alanine aminotransferase (ALT), and total cholesterol were measured using respective commercial kits (GENIA, Seong-Nam, Korea).

### 2.7. Statistical Analyses

Differences between treatments were determined using one-way analyses of variance and multiple comparisons with Bonferroni correction. Statistical significance is reported at *p* < 0.05. All analyses were performed using SPSS Statistics ver. 19.0 (SPSS Inc., Chicago, IL, USA).

## 3. Results

### 3.1. Inhibitory Effects of Hispidulin, p-Synephrine, and p-Octopamine HCl on Adipogenesis in 3T3-L1 Preadipocytes

Before initiating adipogenic differentiation in 3T3-L1 preadipocytes, no cytotoxicity was observed in the presence or absence of hispidulin, *p*-synephrine, and *p*-octopamine HCl using the Ez-Cytox cell viability assay kit. After 24 h of incubation, hispidulin, *p*-synephrine, and *p*-octopamine HCl, at concentrations of 40 µM, each, did not affect the viability of 3T3-L1 preadipocytes (Figure 1A–C). ORO staining showed that treatment with hispidulin and *p*-synephrine inhibited the differentiation of 3T3-L1 preadipocytes into mature adipocytes. However, *p*-octopamine HCl did not inhibit the formation of red-stained lipid droplets, which was, however, slightly inhibited by treatment with 20 µM and 40 µM hispidulin (53.06% ±  2.95% and 43.92% ±  2.54% reduction, respectively). Similarly, formation of red-stained lipid droplets was slightly inhibited by treatment with 20 µM and 40 µM *p-*synephrine (48.37% ±  3.58% and 43.54% ±  0.84% reduction, respectively (Figure 1D–G).

### 3.2. Inhibitory Effects of SOH on Adipogenesis in 3T3-L1 Preadipocytes

As shown in Figure 2A–D, a mixture of *p*-synephrine, *p*-octopamine HCl, and hispidulin, at concentrations of 40 µM, each, did not affect the viability of 3T3-L1 preadipocytes. As shown in Figure 2E–I, co-treatment with 5, 10, 20, and 40 µM hispidulin and 5, 10, 20, and 40 µM *p*-synephrine resulted in stronger inhibition of red-stained lipid droplet formation than treatment with hispidulin or *p*-synephrine alone. Cells treated with equal concentrations of hispidulin and *p*-synephrine (5, 10, 20, or 40 µM, each) showed significant inhibition (53.46% ±  0.98%, 54.08% ±  0.93%, 37.99% ±  0.56%, and 37.38% ±  0.93% reduction, respectively) of the formation of red-stained lipid droplets. Co-treatment with 5, 10, 20, and 40 µM *p*-synephrine and 5, 10, 20, and 40 µM *p*-octopamine HCl resulted in stronger inhibition of red-stained lipid droplet formation than treatment with *p*-octopamine HCl alone. Red-stained lipid droplet formation was slightly inhibited by treatment with equal concentrations (20 and 40 µM, each) of *p*-synephrine and *p*-octopamine HCl (57.91% ±  2.75% and 48.69% ±  0.71% reduction, respectively). Co-treatment with 5, 10, 20, and 40 µM hispidulin and 5, 10, 20, and 40 µM *p*-octopamine HCl resulted in stronger inhibition of red-stained lipid droplet formation than treatment with *p*-octopamine HCl alone. Red-stained lipid droplet formation was slightly inhibited by treatment with equal concentrations (20 and 40 µM, each) of hispidulin and *p*-octopamine HCl (53.33% ±  1.33% and 48.14% ±  0.97% reduction, respectively). SOH treatment resulted in greater inhibition of the formation of red-stained lipid droplets than treatment with hispidulin, *p*-synephrine, or *p*-octopamine HCl alone. Cells treated with equal concentrations of hispidulin, *p*-synephrine, and *p*-octopamine HCl (5, 10, 20, and 40 µM, each) showed significant inhibition (50.38% ±  2.79%, 50.67% ±  1.08%, 40.32% ±  0.87%, and 36.11% ±  1.16% reduction, respectively) of red-stained lipid droplet formation.

### 3.3. Inhibitory Effects of SOH on the Expression of Proteins Involved in Adipogenesis of Differentiated 3T3L-1 Cells

Western blotting was performed to examine the expression of adipogenic marker proteins including PPARγ, C/EBPα, and C/EBPβ in 3T3L-1 cells. Treatment with SOH inhibited the expression of PPARγ, C/EBPα, and C/EBPβ (Figure 3). This suggests that SOH was effective in decreasing adipogenic marker proteins during the eight-day adipocyte differentiation period.

### 3.4. Effects of SOH on Body Weight and Food Intake in HFD-Induced Obese Mice

Body weight of HFD mice was higher than that of mice in the other groups at eight weeks. The weight of HFD mice was 34.7 ± 0.65 g, which was approximately 4.5 g higher than that of the ND control mice. Those mice treated with SOH had a lower body weight (29.1 ± 0.97 g) than HFD mice and ND mice. Mice treated with ATS had a lower body weight (31.9 ± 0.41 g) than the HFD group (Figure 4A). Dietary intake showed a tendency to increase in all groups during the experimental period. The HFD group showed high dietary intake during all experimental periods, except for the first week. The intake of ATS-treated mice showed a similar tendency to the HFD mice without improvement effect. However, SOH-treated mice showed a tendency towards lower intake than HFD mice during all of the experimental period, except the first week. In the last week of the experiment, the food intake of the SOH-treated group was 9.9 ± 0.8 g, which was a approximately 5% decrease in food intake compared with HFD group (10.4 ± 0.78 g) (Figure 4B).

### 3.5. Effects of SOH on Blood Biochemistry in HFD-Induced Obese Mice

Blood creatinine concentrations were 0.31 ± 0.001 mg/dL in the ND group, 0.30 ± 0.001 mg/dL in the HFD group, 0.29 ± 0.001 mg/dL in the ATS group, and 0.32 ± 0.001 mg/dL in the SOH group. The creatinine levels did not differ significantly between groups (Figure 5A). Blood ALT concentrations were 23.20 ± 1.86 IU/L in the HFD group, which showed slightly higher tendency, not statistically significant, than the 18.67 ± 1.76 IU/L in the ND group; it was 18.0 ± 1.54 IU/L in the SOH group, and ALT levels were approximately 22% lower, compared to the HFD group. ALT levels in the ATS group were 37.0 ± 3.69 IU/L, thus significantly higher than in the HFD group, by approximately 59% (Figure 5B). Total blood cholesterol was 191.50 ± 5.45 mg/dL in HFD mice, which was significantly higher than in ND mice (134.33 ± 5.81 mg/dL); it was 148.40 ± 10.68 mg/dL in SOH mice, which was significantly lower than in HFD mice, by approximately 22%. Cholesterol concentrations were 180.80 ± 5.68 mg/dL in ATS mice, which was slightly lower than in HFD mice by approximately 5% (Figure 5C).

## 4. Discussion

In the present study, we assessed the inhibitory effects of SOH on adipogenesis. Underlying molecular mechanisms of SOH in 3T3-L1 cells were assessed by analyzing its effect on the expression of pro-adipogenic proteins. In addition, its anti-obesity effects in an HFD-induced obesity mouse model were evaluated. SOH reduced adipogenesis by inhibiting the expression of pro-adipogenic C/EBPα, C/EBPβ, and PPARγ in 3T3-L1 cells, leading to decreased dietary intake and body weight in mice with HFD-induced obesity. Adipogenesis and lipid accumulation are associated with the development of obesity. A reduction in obesity is associated with the inhibition of adipogenesis and lipid accumulation in adipocytes, and a reduction in the number of adipocytes [20]. Treatment with SOH resulted in stronger inhibition of the formation of red-stained lipid droplets than treatment with hispidulin, *p*-synephrine, or *p*-octopamine HCl alone. In particular, treatment with SOH (5 µM) resulted in slightly stronger inhibition of the formation of red-stained lipid droplets than co-treatment with 5 µM hispidulin and 5 µM *p*-synephrine.

Adipogenesis is regulated by transcription factors such as C/EBPα, C/EBPβ, and PPARγ. During the eight-day adipocyte differentiation period, C/EBPβ is immediately activated and elicits expression of C/EBPα and PPARγ as master regulators. Expression of C/EBPα and PPARγ is vital for the progression of the late stages of adipogenesis. Moreover, PPARγ activation is associated with lipid accumulation [21]. In the present study, treatment with SOH resulted in stronger inhibition of the expression of C/EBPα, C/EBPβ, and PPARγ, compared to co-treatment with hispidulin plus *p*-synephrine. Thus, our results indicate that SOH significantly reduced lipid accumulation and adipogenesis by inhibiting adipogenic transcription factors during the eight-day adipocyte differentiation period.

We used an HFD-induced obesity mouse model to evaluate the anti-obesity activity of SOH. At the end of the experiment, the body weights of HFD mice treated with SOH were significantly lower than those of the HFD controls. Consistent with the decrease in body weight, the SOH-treated group showed a decrease in food intake, compared with the HFD group. These results indicate that the SOH-induced decrease in body weight was due to reduced food intake. We examined changes in the blood levels of creatinine (to assess kidney function), ALT (indicating liver function), and total cholesterol (as an obesity factor), and no changes in blood creatinine levels after the administration of ATS or SOH were observed, suggesting no kidney toxicity. However, blood ALT levels were significantly increased in ATS-treated mice, suggesting liver toxicity. In general, elevated ALT as an indicator of hepatic function is associated with the development of hepatic steatosis [22]. Obesity is a characteristic of metabolic syndrome that causes cardiovascular disease and diabetes due to the accumulation of visceral fat in the body [23]. Recently, normal weight obesity, or skinny obesity, which has high body fat despite normal body weight, has also been known [24]. Total cholesterol level is known as one of the obesity factors because it is higher in obese and/or overweight people compared with normal weight subjects [25]. In our experiment, SOH-treated mice showed a tendency to suppress weight gain and to decrease dietary intake compared to the HFD group. In addition, the levels of total cholesterol and ALT decreased or tended to decrease compared with those in HFD mice.

Thus, our results indicate that SOH treatment may alleviate the development of hepatic steatosis. We demonstrated that SOH plays an important role in adipogenesis and lipid accumulation by inhibiting C/EBPα, C/EBPβ, and PPARγ expression in 3T3-L1 cells. Moreover, SOH administration mitigated HFD-induced body-weight gain, increased dietary intake, and increased the levels of ALT and total cholesterol. Therefore, we suggest that SOH exerted anti-adipogenic effects in 3T3-L1 cells and obesity effects in HFD-fed mice. These results suggest that administration of SOH may be a potential strategy to reduce obesity.

## 5. Conclusions

SOH treatment attenuated the differentiation and lipid accumulation of 3T3-L1 preadipocytes by inhibiting the expression of adipogenic marker proteins, including C/EBPα, C/EBPβ, and PPARγ. In addition, administration of SOH mitigated HFD-induced body-weight gain, increased dietary intake, and increased the levels of ALT and total cholesterol in mice. Our results suggest that anti-obesity effects of SOH result from a decrease in adipogenesis, and that SOH has a positive effect by reducing body-weight gain in obese mice. Thus, administration of SOH may be a promising alternative strategy for the management of obesity-associated metabolic disorders.

## Figures and Tables

**Figure 1 nutrients-14-02164-f001:**
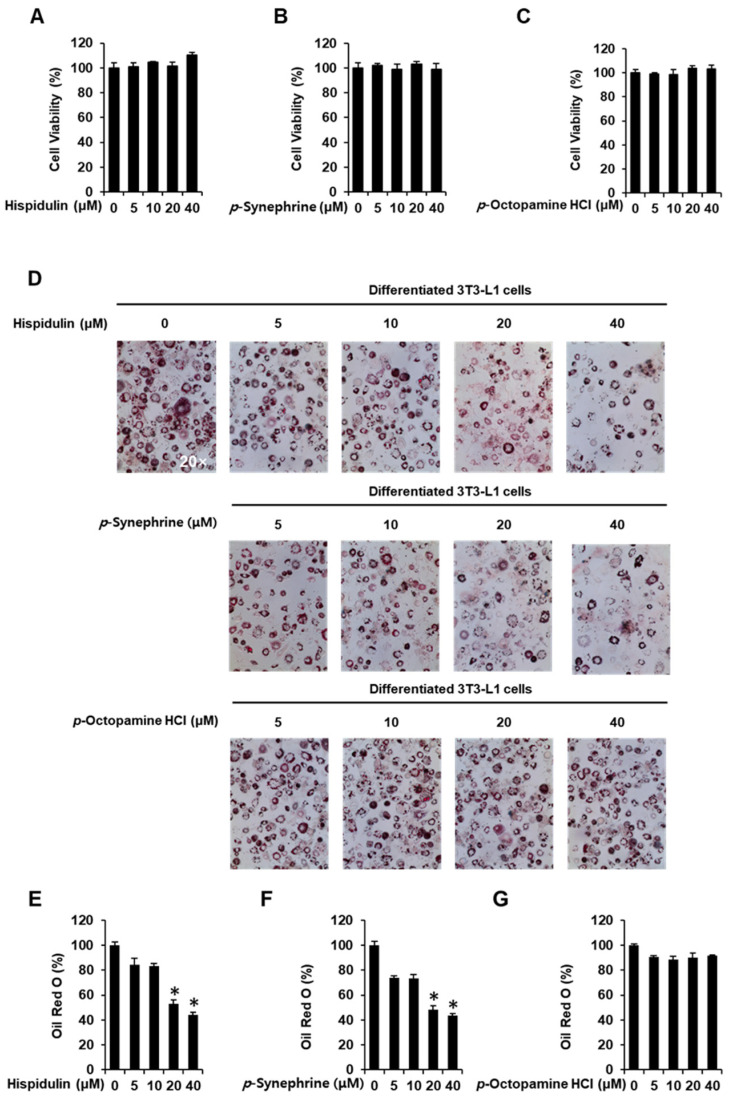
Inhibitory effects of hispidulin, *p*-synephrine, and *p*-octopamine HCl on adipogenesis in 3T3L-1 preadipocytes. Effect of (**A**) hispidulin, (**B**) *p*-synephrine, and (**C**) *p*-octopamine HCl on the viability of 3T3L-1 cells for 8 days using an Ez-Cytox cell viability assay. (**D**) Images of Oil Red O (ORO) staining of differentiated 3T3L-1 cells were produced using an inverted microscope at 20-fold magnification on day 8 after treatment with hispidulin, *p*-synephrine, and *p*-octopamine HCl. (**E**–**G**) Quantification of ORO staining as percentage of untreated control (three independent replicates; * *p* < 0.05, Kruskal–Wallis nonparametric test). Data are shown as means ± standard error of the mean (SEM).

**Figure 2 nutrients-14-02164-f002:**
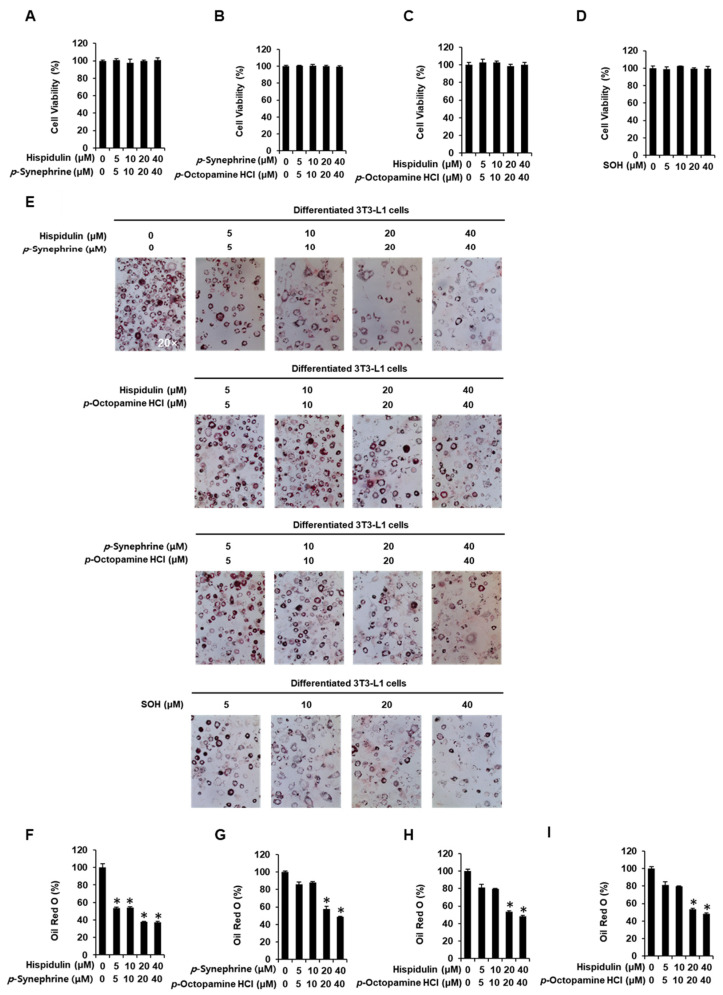
Inhibitory effects of a mixture of *p*-synephrine, *p*-octopamine HCl, and hispidulin (SOH) on adipogenesis of 3T3L-1 preadipocytes. Effect of (**A**) a mixture of hispidulin and *p*-synephrine, (**B**) a mixture of *p*-synephrine and *p*-octopamine HCl, (**C**) a mixture of hispidulin and *p*-octopamine HCl, and (**D**) SOH on the viability of 3T3L-1 cells for 8 days using an Ez-Cytox cell viability assay. (**E**) Images of ORO staining of differentiated 3T3L-1 cells recorded using an inverted microscope at 20-fold magnification on day 8 after treatment with a mixture of hispidulin and *p*-synephrine, a mixture of *p*-synephrine and *p*-octopamine HCl, a mixture of hispidulin and *p*-octopamine HCl, and SOH. (**F**–**I**) Quantification of ORO staining expressed as percentage of untreated controls (three independent replicates; * *p* < 0.05, Kruskal–Wallis nonparametric test). Data are shown as means ± SEM.

**Figure 3 nutrients-14-02164-f003:**
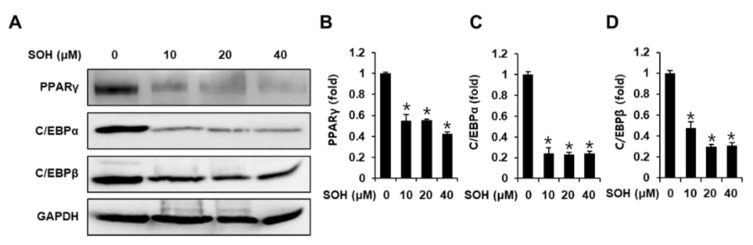
Inhibitory effects of SOH on the expression of adipogenesis-related proteins in differentiated 3T3L-1 cells. (**A**) Protein expression of peroxisome proliferator-activated receptor gamma (PPARγ), CCAAT/enhancer-binding protein alpha (C/EBPα), CCAAT/enhancer-binding protein beta (C/EBPβ), and glyceraldehyde 3-phosphate dehydrogenase (GAPDH) in differentiated 3T3L-1 cells on day 8 after treatment with SOH. (**B**–**D**) Analysis of the ratios of band intensities of PPARγ, C/EBPα, and C/EBPβ in treated cells compared with those of untreated differentiated 3T3L-1 cells (three independent replicates, * *p* < 0.05, Kruskal-Wallis nonparametric test). Data are shown as means ± SEM.

**Figure 4 nutrients-14-02164-f004:**
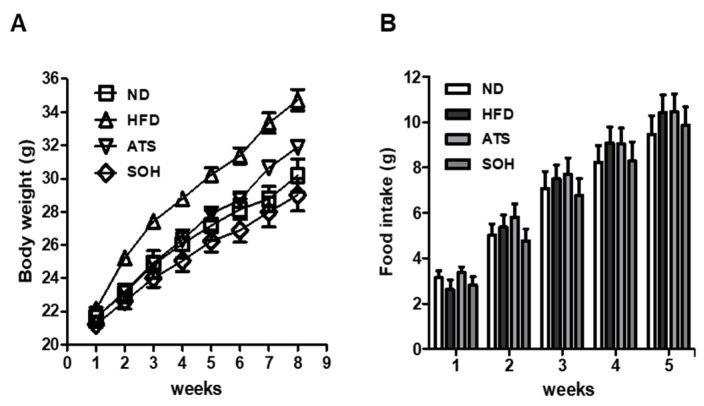
Body weight and food intake. Mice were fed a high-fat diet (HFD) and normal food from one week of age. Body weight and food intake were recorded weekly. (**A**) body weight, (**B**) food intake. ND: normal food control; HFD: control; ATS: HFD + atorvastatin 30 mg/kg oral administration; SOH: HFD + SOH 20 mg/kg (hispidulin, synephrine, and *p*-octopamine HCl in a 1:1:1 mixture), oral administration. Data are shown as means ± SEM. A one-way ANOVA and Bonferroni post-hoc analysis were used to compare normal and HFD, and HFD with each group.

**Figure 5 nutrients-14-02164-f005:**
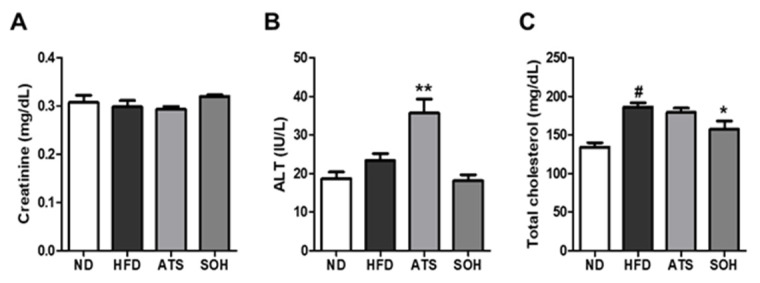
Blood analysis. Blood analysis was performed after killing the mice after 8 weeks. (**A**) Creatinine, (**B**) alanine aminotransferase (ALT), and (**C**) total cholesterol. ND: normal food control, HFD: HFD control; ATS: HFD + atorvastatin 30 mg/kg oral administration; SOH: HFD + SOH 20 mg/kg (hispidulin, synephrine, *p*-octopamine HCl in a 1:1:1 mixture) oral administration. Data are shown as means ± SEM. A one-way ANOVA and Bonferroni post-hoc analysis were used to compare normal and HFD (*^#^ p* < 0.05) and HFD with each group (** p* < 0.05, *** p* < 0.001).

## Data Availability

Data is contained within the article.

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
