# Peer review of "Phytochemical Combination (*p*-Synephrine, *p*-Octopamine Hydrochloride, and Hispidulin) for Improving Obesity in Obese Mice Induced by High-Fat Diet"

_nutrients, 2022, doi:10.3390/nu14102164_

Round 1

Reviewer 1 Report

The main finding of this study is that of p-synephrine, p-octopamine HCl and hispidulin (SOH) supress differentiation of 3T3-L1 preadipocytes into mature adipocytes. Moreover, the Authors report that SOH display anti-obesity effects in  mice fed high fat diet. Nevertheless, manuscript should be deeply revised before consideration it for publication.

  1. Animal study. Diet composition should be descripted in more detailed fashion. It is not enough to state that animals were fed normal of high fat diet. At least % of calories from fat should be provided. How was food intake monitored?
  2. There is no information whether animal study was approved by ethical commission.
  3. Did the Authors study the effects of prolong exposition of 3T3-L1 cells to tested compounds? In this work cytotoxicity was studied in cells treated with hispidulin, p-synephrine, and p-octopamine HCl for 24 h while their effects on adipogenesis were studied after 8 days of treatment. Moreover, the Authors should study the effects of cotreatment on 3T3-L1 viability
  4. Figure 3 shows Western blot data. However, I failed to find description of Western blot in Materials and Methods. This point needs to be descripted. Moreover, molecular weight of tested proteins should be added.
  5. Figure 4. This figure lacks statistical analysis.
  6. Discussion “In the present study, we assessed the inhibitory effects of SOH on adipogenesis, as well as the underlying molecular mechanisms in 3T3-L1 cells..” In my opinion this statement is data overinterpretation. Molecular signalling on antiadipogenic effects of SOH was not elucidated. The authors should try to discuss mechanism by which SOH may downregulate expression of pro-adipogenic proteins.
  7. All figures with images of cells should have scale bares.
  8. List of abbreviations should be added.
  9. “rmp” should be replaced by rcf (g).

Author Response

The main finding of this study is that of p-synephrine, p-octopamine HCl and hispidulin (SOH) supress differentiation of 3T3-L1 preadipocytes into mature adipocytes. Moreover, the Authors report that SOH display anti-obesity effects in mice fed high fat diet. Nevertheless, manuscript should be deeply revised before consideration it for publication.

  1. Animal study. Diet composition should be descripted in more detailed fashion. It is not enough to state that animals were fed normal of high fat diet. At least % of calories from fat should be provided. How was food intake monitored?

Answer: We thank reviewer for valuable comments. We have added content information in “Study Animals and Experimental Design” based on your comments (Lines 136-139).

  1. There is no information whether animal study was approved by ethical commission.

Answer: Changes have made accordingly (Lines 140-145).

  1. Did the Authors study the effects of prolong exposition of 3T3-L1 cells to tested compounds? In this work cytotoxicity was studied in cells treated with hispidulin, p-synephrine, and p-octopamine HCl for 24 h while their effects on adipogenesis were studied after 8 days of treatment. Moreover, the Authors should study the effects of cotreatment on 3T3-L1 viability

Answer: We thank reviewer for helpful comments. We have examined it during the preparation of revision. Hispidulin, p-synephrine, and/or p-octopamine HCl were individually added at different concentrations (5, 10, 20, or 40 μM, each) for 8 days.  As you can see in the Figures 1 and 2, there was no change in 3T3-L1 viability.

  1. Figure 3 shows Western blot data. However, I failed to find description of Western blot in Materials and Methods. This point needs to be descripted. Moreover, molecular weight of tested proteins should be added.

Answer: We apologize for inadvertent error here. We have added the description of Western blot in Materials and Methods including molecular weight of tested proteins (Lines 114-124).

  1. Figure 4. This figure lacks statistical analysis.

Answer: Changes have made accordingly (Lines 261-263).

  1. Discussion “In the present study, we assessed the inhibitory effects of SOH on adipogenesis, as well as the underlying molecular mechanisms in 3T3-L1 cells..” In my opinion this statement is data overinterpretation. Molecular signalling on antiadipogenic effects of SOH was not elucidated. The authors should try to discuss mechanism by which SOH may downregulate expression of pro-adipogenic proteins.

Answer: Thanks for your comments. the content has been modified according to your comments (Lines 290-294).

  1. All figures with images of cells should have scale bares.

Answer: Images of Oil Red O staining of differentiated 3T3L-1 cells were produced using an inverted microscope at 20-fold magnification. Magnification has indicated as suggested.

  1. List of abbreviations should be added.

Answer: Changes have made accordingly (Line 351).

  1. “rmp” should be replaced by rcf (g).

Answer: Changes have made accordingly (Lines 150-151).

Reviewer 2 Report

The study is important and interesting and shows some important results.

However, the manuscript needs to be revised as it has many confusing parts starting with parts of the material and methods that need to be rewritten to be clear, in particular point 2.4. Study Animals and Experimental Design

The results will also have to be better presented. For example in point 3.4 Effects of SOH on Body Weight and Food Intake in HFD-Induced Obese Mice it is said:  “SOH-treated mice howed an approximately 10% decrease in food intake compared with HFD mice. The intake of ATS-treated mice was similar to that of HFD mice (Figure 4B).”

How could they know if the material and methods only say that “The mice had ad libitum access to feed”?  only in fig 4 is it mentioned that mices were fed a high-fat diet (HFD) and body weight and food intake were recorded weekly.

In point 3.5. Effects of SOH on Blood Biochemistry in HFD-Induced Obese Mice it is not correct to say that blood ALT concentrations were 23.20 ± 1.86 IU/L in the HFD group, which was higher than the 18.67 ± 1.76 IU/L in the ND group, because these values are not statistically different, at least not shown in the graph in fig 5.

In graph 5, not all comparisons are made, for example, In graph of total cholesterol not all groups are compared with the ND group.

The discussion needs to be improved.

Can the total cholesterol to be considered an obesity indicator? There are people thin who have high levels of total cholesterol.

Author Response

The study is important and interesting and shows some important results.

However, the manuscript needs to be revised as it has many confusing parts starting with parts of the material and methods that need to be rewritten to be clear, in particular point 2.4. Study Animals and Experimental Design

  1. The results will also have to be better presented. For example in point 3.4 Effects of SOH on Body Weight and Food Intake in HFD-Induced Obese Miceit is said:  “SOH-treated mice howed an approximately 10% decrease in food intake compared with HFD mice. The intake of ATS-treated mice was similar to that of HFD mice (Figure 4B).”

Answer: We thanks reviewer for helpful comments. Changes have made accordingly (Lines 248-254).

  1. How could they know if the material and methods only say that “The mice had ad libitum access to feed”?  only in fig 4 is it mentioned that mice were fed a high-fat diet (HFD) and body weight and food intake were recorded weekly.

Answer: We apologize for insufficient explain of materials and methods. Detailed information has added accordingly (Lines 136-139).

  1. In point 3.5. Effects of SOH on Blood Biochemistry in HFD-Induced Obese Miceit is not correct to say that blood ALT concentrations were 23.20 ± 1.86 IU/L in the HFD group, which was higher than the 18.67 ± 1.76 IU/L in the ND group, because these values are not statistically different, at least not shown in the graph in fig 5.

Answer: “Not significant changes” has mentioned in the revision following reviewer’s indication (Lines 269-272).

  1. In graph 5, not all comparisons are made, for example, In graph of total cholesterol not all groups are compared with the ND group.

Answer: We apologize for this inadvertent error here. Changes have made accordingly (Lines 286-287).

  1. The discussion needs to be improved.

Answer: We thank reviewer for helpful comments. We have added more discussion (Lines 290-294, 323-331).

  1. Can the total cholesterol to be considered an obesity indicator? There are people thin who have high levels of total cholesterol.

Answer: We have studied your comments and added related explanation and reference accordingly (Lines 323 - 331).

Reviewer 3 Report

This is a good article in which the authors described the effect of the phytochemical combination of p-synephrine, p-octopamine hydrochloride, and ispidulin in reducing adipogenesis in 3T3-L1 adipocytes and body weight in animals fed on HFD. The methods used are not very innovative and the results presented are few and preliminary to say that the use of these compounds can improve obesity induced by a hyperlipidic diet.

Regarding the experimental design, the authors should specify the way in which the animals were sacrificed, avoiding the use of the word “killing”.The blood was collected after the animals were sacrificed. The authors should specify whether biochemical analyzes were performed on serum or plasma.

The authors performing a western blot analysis to examine the expression of adipogenic marker proteins such as PPARγ, C/EBPα AND C/EBPβ in 3T3-L1 cells treated with SOH suggesting that this mixture was effective in decreasing adipogenic marker proteins expression during the eight-day adipocyte differentiation period. Can the authors measure the protein levels of adipogenic markers of 3T3-L1 cells treated individually with p-synephrine, p-octopamine HCl, and hispidulin?

The authors evaluate the effect of SOH on blood biochemistry in HFD induced Obese mice such as Creatinine, ALT and total cholesterol. Can the authors also measure the levels of triglycerides, HDL and LDL cholesterol ? It would be interesting to evaluate serum levels of leptin and adiponectin in serum of animals.

Treatment with SOH resulted in stronger inhibition of the formation of red-stained lipid droplets than treatmentwith hispidulin, p-synephrine, or p-octopamine HCl alone. It would be interesting to evaluate in subcutaneous and visceral white adipose tissue of N, HFD, ATS SOH animals the protein levels of lipid droplet-associated protein, Perilipin 1 (PLIN1) and lipases. This step is essential to evaluate the inhibition of lipid accumulation in adipocytes of animals treated with ATS and SOH mixture. It is also strongly recommended to evaluate the accumulation of visceral fat and the morphology of visceral and subcutaneous WAT in the ATS and SOH group.

The authors suggesting that treatment with SOH alleviates the development of hepatic steatosis. The authors should provide evidence that there is an improvement in the development of fatty liver disease (NAFLD) measuring the Hepatic content of glycogen, ceramide and triglyceride in ATS and SOH animals.

Involvement of the adrenergic nervous system is essential in the activation of brown adipose tissue and in browning. Considering that p-octopamine is a adrenergic agonist, is it possible that there is an activation of the browning process in the subcutaneous and visceral white adipose tissue of ATS and SOH animals ? It is strongly recommended to measure the protein levels of some browning markers, for example PRDM16, UCP1 and C/EBPβ in subcutaneous and visceral white adipose tissue of ATS and SOH animals.

Author Response

This is a good article in which the authors described the effect of the phytochemical combination of p-synephrine, p-octopamine hydrochloride, and ispidulin in reducing adipogenesis in 3T3-L1 adipocytes and body weight in animals fed on HFD. The methods used are not very innovative and the results presented are few and preliminary to say that the use of these compounds can improve obesity induced by a hyperlipidic diet.

  1. Regarding the experimental design, the authors should specify the way in which the animals were sacrificed, avoiding the use of the word “killing”.The blood was collected after the animals were sacrificed. The authors should specify whether biochemical analyzes were performed on serum or plasma.

Answer: We thank reviewer for helpful comments. “Study Animals and Experimental Design” section has corrected according to the comments (Lines 140 - 145).

  1. The authors performing a western blot analysis to examine the expression of adipogenic marker proteins such as PPARγ, C/EBPα AND C/EBPβ in 3T3-L1 cells treated with SOH suggesting that this mixture was effective in decreasing adipogenic marker proteins expression during the eight-day adipocyte differentiation period. Can the authors measure the protein levels of adipogenic markers of 3T3-L1 cells treated individually with p-synephrine, p-octopamine HCl, and hispidulin?

Answer: In the previous our study (Ref. 11), the protein levels of adipogenic markers of 3T3-L1 cells treated individually with p-synephrine and hispidulin have already evaluated. During the preparation of revision, we examined the protein levels of adipogenic markers of 3T3-L1 cells treated with p-octopamine HCl. As you can see in the Figures S1, there was no change in the protein levels of adipogenic markers.

3. The authors evaluate the effect of SOH on blood biochemistry in HFD induced Obese mice such as Creatinine, ALT and total cholesterol. Can the authors also measure the levels of triglycerides, HDL and LDL cholesterol ? It would be interesting to evaluate serum levels of leptin and adiponectin in serum of animals.

 Answer: We thank reviewer for thorough review. Further in vivo analysis is not possible because other lipid markers were not considered in our experimental design. However, based on the comments of reviewers, we’re planning to conduct additional research for longer period in the near future.

4. Treatment with SOH resulted in stronger inhibition of the formation of red-stained lipid droplets than treatmentwith hispidulin, p-synephrine, or p-octopamine HCl alone. It would be interesting to evaluate in subcutaneous and visceral white adipose tissue of N, HFD, ATS SOH animals the protein levels of lipid droplet-associated protein, Perilipin 1 (PLIN1) and lipases. This step is essential to evaluate the inhibition of lipid accumulation in adipocytes of animals treated with ATS and SOH mixture. It is also strongly recommended to evaluate the accumulation of visceral fat and the morphology of visceral and subcutaneous WAT in the ATS and SOH group.

Answer: We thank reviewer for thorough review. As answered to above question #4, further in vivo analysis is not possible because other lipid markers were not considered in our experimental design. However, based on the comments of reviewers, we’re planning to conduct additional research for longer period in the near future.

5. The authors suggesting that treatment with SOH alleviates the development of hepatic steatosis. The authors should provide evidence that there is an improvement in the development of fatty liver disease (NAFLD) measuring the Hepatic content of glycogen, ceramide and triglyceride in ATS and SOH animals.

 Answer: We’ve studied reviewer’s comments and got to know that our experimental condition is short and not suitable for NAFLD, and have been deleted contents of steatosis in discussion.

  1. Involvement of the adrenergic nervous system is essential in the activation of brown adipose tissue and in browning. Considering that p-octopamine is a adrenergic agonist, is it possible that there is an activation of the browning process in the subcutaneous and visceral white adipose tissue of ATS and SOH animals? It is strongly recommended to measure the protein levels of some browning markers, for example PRDM16, UCP1 and C/EBPβ in subcutaneous and visceral white adipose tissue of ATS and SOH animals.

Answer: We thank reviewer for thorough review. As answered to aforementioned questions, further in vivo analysis is not possible because other lipid markers were not considered in our experimental design. However, based on the comments of reviewers, we’re planning to conduct additional research for longer period in the near future.

Round 2

Reviewer 1 Report

I have no more comments to Authors.

Reviewer 2 Report

Some subtitles still need to be revised

For example in Fig 4 it makes no sense to be in the caption “Mice were fed a high-fat diet (HFD) and normal food from one week of age. Body weight and food intake were recorded weekly” as they are information that must be included in the material and methods chapter. The caption of the same figure appears “A one-way ANOVA and Bonferroni post-hoc analysis were used to compare normal and HFD (# p < 0.05) and HFD with each group (* p < 0.05, **p < 0.001)”, but in reality on the graph there are no * and **

Reviewer 3 Report

I thank the authors for their answers, but I consider the work unsuitable for publication in Nutrients, as no additional experiments that we requested have been carried out.